# A Fenton-Like Nanocatalyst Based on Easily Separated Magnetic Nanorings for Oxidation and Degradation of Dye Pollutant

**DOI:** 10.3390/ma13020332

**Published:** 2020-01-11

**Authors:** Xiaonan Li, Jinghua Li, Weilu Shi, Jianfeng Bao, Xianyuan Yang

**Affiliations:** School of Medical Technology and Engineering, Henan University of Science and Technology, Luoyang 471023, China; 15638744120@163.com (X.L.); L13782166014@163.com (W.S.); baoguojianfeng@gmail.com (J.B.); yxy820829@163.com (X.Y.)

**Keywords:** magnetic, Fenton, crystal structure, dyes degradation, recycling

## Abstract

In this study, uniform Fe_3_O_4_ magnetic nanorings (Fe_3_O_4_-MNRs) were prepared through a simple hydrothermal method. The morphology, magnetic properties, and structure of the product were characterized by transmission electron microscope (TEM), scanning electron microscope (SEM), high resolution transmission electron microscopy (HRTEM), vibrating sample magnetometer (VSM), X-ray powder diffraction (XRD), and X-ray photoelectron spectroscopy (XPS), respectively. The Fe_3_O_4_-MNRs were used as Fenton-like catalysts in the presence of hydrogen peroxide (H_2_O_2_) and showed excellent Fenton-catalytic activity for degradation of organic dyes such as Methylene blue (MB), Rhodamine B (RhB), and Bromophenol blue (BPB). Furthermore, the obtained Fe_3_O_4_-MNRs could be recycled after used for several times and still remained in a relative high activity and could rapidly be separated from the reaction medium using a magnet without considerable loss. All results reveal that Fe_3_O_4_-MNRs have potential for the treatment of dyes pollutants.

## 1. Introduction

With the advancement of technology and society, dyes are extensively used in textiles, plastics, leather, pharmaceuticals, food, cosmetics, dyestuffs, tanning, and printing industries [1,2]. Considerable amount of colored wastewater is generated and may cause damage to aquatic ecosystems or even bring about serious risks to human health [3]. Therefore, there is an urgent and serious demand for efficient treatment of these discharged dyes. A great many physical and chemical methods have been exploited to remove/degrade those dyes from wastewater, such as physical adsorption, precipitation coagulation, flocculation, reverse osmosis, filtration, membrane separation, biological process, chemical oxidation, and catalytic degradation [4,5,6,7,8,9]. Nevertheless, some chemical and physical methods such as adsorption and coagulation only transfer pollutants and require more processing.

Since very recently, it has been more appropriate to utilize the advanced oxidation processes (AOPs) for organic pollutant treatment, including photolysis, photocatalysis [10], Fenton process [11,12], ozonation [13,14], and sonolysis [15]. Typically, the Fenton reaction is one of the most explored methods among AOPs for degradation of dyes to harmless products, which uses hydrogen peroxide as an oxidant. With the assistance of H_2_O_2_, the Fenton agents could generate hydroxyl radicals (•OH) automatically and thus degrade organic matters to CO_2_ and H_2_O ultimately [16]. To this end, tremendous efforts have been focused on using alternative catalysts for wastewater treatment such as titanium dioxide (TiO_2_) [17], cadmium sulphide (CdS) [18], zinc oxide (ZnO) [19], manganese dioxide (MnO_2_) [20], copper oxide [21], tungsten trioxide (WO_3_) [22], and iron oxides [23,24], which can effectively degrade organic contaminants by a typical Fenton procedure; the Fenton-like oxidation reactions have been commonly used to produce free radicals via catalyzing H_2_O_2_ with ferrous ions (Fe^2+^/Fe^3+^) in acidic media [25,26]. Typically, the iron oxides nanomaterials could be used as an oxidase to catalytically convert H_2_O_2_ to •OH. In many previous studies, the generally accepted mechanism of •OH generated in Fenton reaction includes a series of cyclic reactions, which can be described by the classical Haber–Weiss cycle (Equations (1)–(7)) [27].
Fe^2+^ + H_2_O_2_ → Fe^3+^ + •OH + OH^−^
*k* = (63 − 70) mol^−1^∙L∙s^−1^(1)
Fe^3+^ + H_2_O_2_ → Fe^2+^ + HO_2_• + H^+^
*k* = (0.001 − 0.01) mol^−1^∙L∙s^−1^(2)
•OH + H_2_O_2_ → HO_2_• + H_2_O
*k* = 3.3 × 10^7^ mol^−1^∙L∙s^−1^(3)
•OH + Fe^2+^ → Fe^3+^ + OH^−^
*k* = 3.2 × 10^8^ mol^−1^∙L∙s^−1^(4)
Fe^2+^ + HO_2_• + H^+^ → Fe^3+^ + H_2_O_2_
*k* = (1.2 − 1.3) × 10^6^ mol^−1^∙L∙s^−1^(5)
Fe^3+^ + HO_2_• → Fe^2+^ + O_2_ + H^+^
*k* = (1.3 − 2.0) × 10^3^ mol^−1^∙L∙s^−1^(6)
2HO_2_• → H_2_O_2_ + O_2_
*k* = 8.3 × 10^5^ mol^−1^∙L∙s^−1^(7)

However, for most commonly used Fenton catalysts, there are still some limitations due to low catalytic efficiency within inapposite pH ranges and difficulty in recycling usage. Meanwhile, the crystallization, morphology, and specific surface area of the catalyst are also the main factors that determine the Fenton reaction rate [28].

Herein, we processed a simple and feasible one-step hydrothermal method for synthesis of hollow Fe_3_O_4_ magnetic nanorings (Fe_3_O_4_-MNRs). Typically, Fe_3_O_4_-based nanomaterials have been shown to be a promising Fenton candidate for the dyes degradation due to the Fe^2+^/Fe^3+^ Fenton system with the assistance of H_2_O_2_. Meanwhile, the magnetic nanomaterials could be easily separated from the reaction medium by using an external magnetic field and for the recycling usage. Various characterizations indicate that Fe_3_O_4_-MNRs have extremely small size and high surface-area to volume ratio. In this study, three typical dyes, Methylene blue (MB), Rhodamine B (RhB), and Bromophenol blue (BPB), were selected as the model waste dyes to estimate the catalytic effect of the Fe_3_O_4_-MNRs Fenton-like agents on the decomposition of organic contaminants. The Fe_3_O_4_-MNRs present high catalytic efficiency for various dyes. Moreover, it can be easily recovered with an external magnet and reused after simple washing. The Scheme 1 shows the dyes degradation mechanism of Fe_3_O_4_-MNRs with assistance of H_2_O_2_.

## 2. Materials and Methods

### 2.1. Reagents and Materials

Iron chloride (FeCl_3_, 99%), ammonium dihydrogenophosphate (NH_3_H_2_PO_4_, 99%), Methylene blue (MB, 3,7-Bis(dimethylamino)-5-phenothiazinium chloride, dye content ≥98%), Rhodamine B (RhB, 9-(2-carboxyphenyl)-3,6-bis (diethylamino) xanthylium chloride, dye content ≥98%), Bromophenol blue (BPB, 4,4ʹ-(1,1-Dioxido-3H-2,1-benzoxathiol-3-ylidene) bis [2,6-dibromo-phenol], dye content ≥98%), anhydrous sodium sulfate (Na_2_SO_4_, 99%) and other conventional reagents were all bought from Aladdin Co., Ltd. (Shanghai, China) without any further purification.

### 2.2. Synthesis of Fe_3_O_4_-MNRs

Firstly, iron oxide nanorings (α-Fe_2_O_3_-NRs) were synthesized via a hydrothermal treatment [29]. FeCl_3_ (0.02 M), NH_3_H_2_PO_4_ (0.20 mM) and Na_2_SO_4_ (0.56 mM) were dissolved into 80 mL distilled water. The precursor solution was stirred at room temperature for 30 min. After the mixtures dissolved completely, the obtained solution was transferred into a Teflon-lined stainless-steel autoclave. The reaction was performed at 220 °C for 12 h. The resulting precipitate particles were separated by centrifugation and washed with ethanol for eight times. Then the brownish yellow products were dried in a vacuum drying oven at 60 °C for 8 h and noted as α-Fe_2_O_3_-NRs. Then, the Fe_3_O_4_-MNRs were prepared via a reduction process with α-Fe_2_O_3_-NRs. Typically, the dried α-Fe_2_O_3_-NRs were annealed in a furnace at 380 °C under a continuous hydrogen/argon (*v/v*: H_2_/Ar = 5%) gas flow for 6 h. The resulting Fe_3_O_4_-MNRs was collected with a magnet and washed with ethanol and distilled water each for eight times. To prepare type I Fe_3_O_4_-MNPs, briefly, FeCl_3_ (0.02 M) and NH_3_H_2_PO_4_ (0.40 mM) were dissolved into 80 mL distilled water, then the obtained solution was transferred into a Teflon-lined stainless-steel autoclave and reacted at 220 °C for 12 h. The resulting precipitate particles were separated by centrifugation and washed with ethanol eight times. Then the precipitates were annealed in a furnace at 380 °C under a continuous hydrogen/argon (*v/v*: H_2_/Ar = 5%) gas flow for 6 h. The preparation of Type II Fe_3_O_4_-MNPs was the same as Type I Fe_3_O_4_-MNPs; the only difference was the concentration of the reactive solutes (FeCl_3_, 0.02 M and Na_2_SO_4_, 0.56 mM).

### 2.3. Characterization

The morphological of as-synthesized α-Fe_2_O_3_-NRs and Fe_3_O_4_-MNRs samples were observed by Scanning Electron Microscope (SEM, JEOL JSM-7800F, Tokyo, Japan), transmission electron microscopy (TEM) and high-resolution transmission electron microscopy (HRTEM, JEOL JEM-2100, Tokyo, Japan). The magnetization values of α-Fe_2_O_3_-NRs and Fe_3_O_4_-MNRs were measured by the vibrating sample magnetometer (VSM, Lake Shore 7410, Carson, CA, USA). The XRD patterns of Fe_3_O_4_-MNRs sample were generated on a X-ray diffraction diffractometer (D8 Advance, Bruker, Billerica, MA, USA) with Cu Kα radiation (λ = 1.5147 Å). The chemical composition of the as-prepared Fe_3_O_4_-MNRs was characreized by X-ray photoelectron spectroscopy (XPS, Kratos Axis Ultra DLD, Kyoto, Japan). The transmittance spectra of dyes degradation samples were measured using UV-Vis spectrophotometer (Thermo Scientific NanoDrop One, Waltham, MA, USA). Spectra were recorded at room temperature in steps of 0.5 nm in the range 190–850 nm. The free radicals triggered by Fe_3_O_4_-MNRs in H_2_O_2_ were recorded with an electron paramagnetic resonance (EPR) spectroscope (Bruker EMXnano, Karlsruhe, Germany).

### 2.4. Free Radical (•OH) Generation and Detection

For detection of free radicals in the Fenton-like catalysis, EPR was used to record signals of spin adducts that were produced by active free radicals reacting with 5,5-dimethyl-1-pyrroline-*N*-oxide (DMPO) [30]. Briefly, BMPO was used as a scavenger to capture •OH to form relatively stable adducts DMPO-•OH, and the types of free radical were distinguished according to the EPR spectra of the adducts. Typically, Fe_3_O_4_-MNRs (1.0 g∙L^−1^), BMPO (40 mM), and dyes (RhB, MB, and BPB, respectively, 0.2 mg∙mL^−1^) were mixed together for 4 min. Each sample was then moved into a capillary and sealed. The capillary was put into a quartz tube, which was inserted in the EPR cavity. Then the 1:2:2:1 multiplicity characteristic peaks of DMPO-•OH adducts were recorded by EPR immediately. The EPR parameters for all samples were as follows: center field = 3510 G, sweep width = 100 G, microwave frequency = 9.85 GHz, microwave power = 6.325 mW, modulation frequency = 100 kHz.

### 2.5. Catalytic Activity Measurements

The synthesized Fe_3_O_4_-MNRs were evaluated through the catalytic oxidation of dyes in presence of H_2_O_2_. In brief, effects of the temperature and pH tolerances on the residual activity of Fe_3_O_4_-MNRs were tested in buffer solutions at pH 5.0, temperature from 20 to 90 °C and in buffer solutions at 20 °C within the pH range from 3.0 to 11.0 for 2 h, respectively. Influence of reaction conditions such as catalysis reaction time and initial dyes concentration on dye degradation efficiency was also investigated. The concentrations of dyes in the assay solutions were measured by the method reported by previous reports [31,32,33].

Typically, Fe_3_O_4_-MNRs (6.0 mg) was firstly dispersed into three dyes (MB, BPB, and RhB) solution (0.2 mg∙mL^−1^, 1.0 mL) containing 1.0 mL H_2_O_2_ at room temperature. Then, 1 µL sample was drawn by a sampler and measured in time course by monitoring the absorbance change at 554 nm for RhB, 665 nm for MB, and 440 nm for BPB with a UV-vis spectrophotometer, respectively. The amount of samples was so small that it could be negligibly compared to the experimental sample. Meanwhile, blank measurements were taken under the same operated condition but without addition of Fe_3_O_4_-MNRs.

### 2.6. The Degradation Mechanism

The products in the reaction solution during the degradation were measured using gas chromatography–mass spectrometry (GC-MS). The intermediates were examined with GC-MS (Agilent 6890 N) equipped with a fused silica capillary column (DB-5, 60 m long, 0.32 mm i.d.) The column temperature was raised from 40 °C to 260 °C at 5 °C/min during the measurement. The production of CO_2_ was monitored by a Carbon dioxide gas analyzer (Thermo Scientific™ 410i, Waltham, MA, USA).

### 2.7. Statistical Analysis

Statistical significance was determined using an analysis of variance and Tukey’s test (OriginPro, version 8.0, OriginLab Corporation, Northampton, MA, USA). Statistical significance was established at *p* < 0.05.

## 3. Results

### 3.1. Morphology

The morphology of as-synthesized α-Fe_2_O_3_-NRs and Fe_3_O_4_-MNRs was evaluated by TEM (Figure 1a,b) and SEM (Figure 1c,d). It showed that the Fe_3_O_4_-MRs presented homogeneous shape and relatively uniform size. There was almost no change in the size on conversion from α-Fe_2_O_3_ to Fe_3_O_4_ nanorings. Meanwhile, the SEM images of the type I and type II Fe_3_O_4_ nanoparticles were shown in Appendix A. Figure 1e,f shows HRTEM images of the obtained α-Fe_2_O_3_-NRs and Fe_3_O_4_-MNRs, respectively. Massive step and terrace atoms oriented along the zone axison were obviously observed [29]. In Figure 1e, the interplanar spacing of 0.250 nm for α-Fe_2_O_3_ could be obtained. The selected area electron diffraction (SAED) patterns with dot-matrix shape in the inset indicate that the α-Fe_2_O_3_ was a single crystal structure. In Figure 1f, the HRTEM image of Fe_3_O_4_-MNRs shows the characteristic lattice fringe of 0.290 nm and 0.480 nm, indicating the formation of Fe_3_O_4_-MNRs. Meanwhile, the microcosmic structure characterization of synthesized Fe_3_O_4_-MNRs could be confirmed by the SAED patterns in the inset which shows the dot matrix shape. In addition, the Fe_3_O_4_-MNRs possessed high specific surface area (about 109.3 m^2^/g, Appendix A) with Brunauer-Emmett-Teller (BET) analysis, which was important for catalytic reaction.

### 3.2. Size Distributions

The inner ring average diameter of the Fe_3_O_4_-MNRs was determined to be about 75.2 nm. The outer ring average diameter of the Fe_3_O_4_-MNRs was determined to be about 155.6 nm (Figure 2, *n* = 300). The thickness of ring wall should be about 40.2 nm. Meanwhile, the average crystallite size can be computed by Scherrer equation; the correlation calculation results were listed in Appendix A; the crystallite sizes of the Fe_3_O_4_-MNRs were about 22 to 35 nm, which were near to the result of TEM. Furthermore, the average diameter of the type I and type II Fe_3_O_4_ nanoparticles were determined to be about 337.49 (*n* = 300, Appendix A) and 196.52 nm (*n* = 300, Appendix A), respectively.

### 3.3. Magnetization

The hysteresis loops of the as-prepared α-Fe_2_O_3_-NRs and Fe_3_O_4_-MNRs were recorded at room temperature (Figure 3). The saturation magnetization of the material was about 62.4 emu/g, which showed that the Fe_3_O_4_-MNRs emerged good magnetic properties and can be easily separated from the solution with an external magnet.

### 3.4. X-ray Diffraction

Then composition of the obtained nanomaterials was further detected by X-ray diffraction (XRD) analysis (Figure 4). Figure 4 shows typical XRD patterns of the Fe_3_O_4_-MNRs; the result indicated that all of the characteristic peaks (30.1°, 35.5°, 43.1°, 53.4°, and 57.0°) matched well with pure spinel Fe_3_O_4_ (JCPDS no. 19–0629) [29].

### 3.5. X-ray Photoelectron Spectroscopy

The element composition and status of Fe_3_O_4_-MNRs was measured using X-ray photoelectron spectroscopy (XPS). Typically, the Fe, O, and C elements were observed in XPS spectra of Fe_3_O_4_-MNRs (Figure 5). In the inset of Figure 5, the peaks at 711.1 eV and 724.6 eV were assigned to the 2p_3/2_ and 2p_1/2_ of Fe, the peak at 530.1 eV was attributed to the 1s of O, and the peak at 285.2 eV was assigned to the C 1s, which were consistent with the previous literature [34]. The above results confirmed the fact that Fe_3_O_4_-MNRs were successfully synthesized.

### 3.6. Hydroxyl Radicals (•OH) Generation and Monitoring

The amount of •OH was determined by using DMPO, which is the •OH specific spin-trapping agent. Accordingly, the Fe_3_O_4_-MNRs promoted the production of free radical •OH with the assistance of H_2_O_2_; in contrast, the radicals signal was not obvious in H_2_O_2_ or Fe_3_O_4_-MNRs group alone (Figure 6). The results indicated that the Fe_3_O_4_-MNRs could generate •OH effectively, which was ascribed to the Fenton-like effect between the iron ions (release from Fe_3_O_4_-MNRs) and H_2_O_2_. Then the produced •OH could be used for the subsequent dye decoloration treatment [35].

### 3.7. Dye Degradation through a Catalytic Fenton-Like Reaction

The quantitative degradation ability of the Fe_3_O_4_-MNRs to MB, BPB and RhB solution were examined by UV-vis. The degradation spectra of the MB solution in Fe_3_O_4_-MNRs/H_2_O_2_ system during the Fenton catalytic decoloration are shown in Figure 7a. The absorbance peaks had a regular reduction with time and almost completely disappeared after 2 days, suggesting that MB was effectively degraded. Next, the spectra of the RhB and BPB solution were also recorded from 30 min to 240 min, as shown in Figure 7b,c and Appendix A; compared to the blank control, the Fe_3_O_4_-MNRs also showed good degradation performance for different kinds of dyes. Furthermore, optical images (Figure 7d) of the dye degradation were monitored during the treatment periods. Accordingly, with the assistance of H_2_O_2_, the Fe_3_O_4_-MNRs exhibited excellent Fenton catalytic efficiency for dye decoloration.

### 3.8. Dye Degradation Study

To confirm the Fe_3_O_4_-MNRs could degrade wastewater efficiently, we performed the experiments of dye decoloration at different treatment conditions, such as pH, temperature, and initial dye concentration. Briefly, the degradation efficiency of Fe_3_O_4_-MNRs demonstrated a pH dependence from 3.0 to 11.0; the maximum catalytic efficiency occurred around pH 4.0–6.0 (Appendix A); this phenomenon was related to the fact that the iron metallic ion leaches in acid pH value more effectively. The temperature-dependent performances were also researched over increasing temperatures from 4 to 60 °C, the Fe_3_O_4_-MNRs showed excellent removal efficiency (above 70%) at different temperature ranges (Appendix A). The initial concentration of dyes highly affected the removal efficiency. Around 90% of dyes were degraded by Fe_3_O_4_-MNRs at low concentrations of 0.1 mg∙mL^−1^, in comparison with around 70% removal efficiency for a high concentration of 1.0 mg∙mL^−1^ (Appendix A). This result could be interpreted as follows: The dye removal capacity shows a concentration-dependent behavior.

Meanwhile, the Fe_3_O_4_-MNRs presented time-dependent degradation behavior for all the three dyes in the treating processes; the colored pollutant could be eliminated completely after 2 days treatment period (Figure 8 and Appendix A). Furthermore, dyes’ (MB, RhB, and BPB) degradation efficiency by different Fe_3_O_4_ nanoparticles (240 min, 20 °C, pH 5.0) were shown in Appendix A. The Fe_3_O_4_-MNRs exhibited better degradation effect than Type I and Type II Fe_3_O_4_ nanoparticles and exhibited significant difference, probably due to its cavity structure and larger specific surface area. In addition, there was no significant difference in catalytic performance of the three catalysts after 10 cycles, which all showed excellent reusability (Appendix A). Taken together, the as-prepared Fe_3_O_4_-MNRs exhibited more excellent degradation performance for various colored pollutants compared to the other two catalysts.

### 3.9. Regeneration of the Catalyst

In general, the reuse of a degradation nanomaterial is a very important characteristic once these can be regenerated and reused in another wastewater treatment processes. Figure 9 shows the results obtained from the reuse cycles of Fe_3_O_4_-MNRs in the Fenton degradation of the dyes solution; the Fe_3_O_4_-MNRs can retain above 85% of its initial activity after being reused for 10 cycles. Furthermore, the Fe_3_O_4_-MNRs after 10 cycles were characterized by SEM (Appendix A), particle size distribution (Appendix A), XRD pattern (Appendix A), and XPS spectra (Appendix A). The result showed that the morphology and composition had no significant difference from the raw Fe_3_O_4_-MNRs.

### 3.10. The Degradation Mechanism

In order to confirm the process of dyes oxidized degradation, the degradation process was monitored by a GC-MS method. Typically, after 60 min degradation period, for the MB solution, some small organic molecules such as methylbenzene, styrene, phenol, p-nitrophenol, trimethylphenol, and 4-Aminothiophenol, N,N,S-trimethyl could be detected (Figure 10a). For the RhB solution, small organic molecules such as methylbenzene, phenol, p-nitrophenol, and trimethylphenol could be detected (Figure 10b). For the BPB solution, some small organic molecules such as tetrahydro-2,5-dimethyl, methylbenzene, phenol, and trimethylphenol could be detected (Figure 10c). The kinetics of dyes disappearance are given in Appendix A. It can be seen that the quantities of dye degradation increased with time. During the 240 min degradation process, the organic molecules were degraded as CO_2_ (Appendix A) and inorganic ions (Figure 10d–f). For the MB solution, the final ionic products are SO_4_^2−^, NH_4_^+^, and NO_3_^−^, respectively. For the RhB solution, the final ionic products are NH_4_^+^ and NO_3_^−^. For the BPB solution, the final ionic products are Br^−^ and SO_4_^2−^. Accordingly, the dyes were degraded.

## 4. Conclusions

In summary, α-Fe_2_O_3_-NRs and Fe_3_O_4_-MNRs were successfully prepared. The Fe_3_O_4_-MNRs exhibited an extraordinarily high catalytic activity for dye degradation in wastewater at room temperature. Surprisingly, EPR measurement combining with scavenger experiment indicated that Fenton-like catalysis contributed to dye decoloration. The range of pH values for Fenton-like oxidation was extended, and the leaching of iron from the synthesized Fe_3_O_4_-MNRs after degradation was found to be negligible and therefore could overcome the shortcoming of traditional Fenton reaction in a wide pH range. Moreover, the Fe_3_O_4_-MNRs could be recycled easily from solution by an external magnet. All results displayed that the Fe_3_O_4_-MNRs were an excellent catalyst with great potential for all kinds of wastewater treatment. However, understanding of the chemical degradation mechanism for different intermediate products is limited and warrants continued intensive study. Next, we will investigate the possible toxicity and complete degradation of these intermediates in our further research.

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
