# Peer review of "A Fenton-Like Nanocatalyst Based on Easily Separated Magnetic Nanorings for Oxidation and Degradation of Dye Pollutant"

_materials, 2020, doi:10.3390/ma13020332_

Round 1

Reviewer 1 Report

In this study, Fe3O4 nanorings were prepared hydrothermally and used for the Fenton reaction. The synthesized Fe3O4 nanorings are characterized by various methods including (HR) TEM, SEM, VSM, XRD, and XPS, and methylene blue, rhodamine B, and bromophenol blue are decomposed by Fenton reaction as model organic dyes.

The quality of each experimental result is generally good, but there is little explanation or discussion about the results. Therefore, the reviewer recommends the authors to revise the manuscript and refer to the literature to add a more detailed explanation of the results.

Author Response

Comments to the Author

In this study, Fe3O4 nanorings were prepared hydrothermally and used for the Fenton reaction. The synthesized Fe3O4 nanorings are characterized by various methods including (HR) TEM, SEM, VSM, XRD, and XPS, and methylene blue, rhodamine B, and bromophenol blue are decomposed by Fenton reaction as model organic dyes.

Response: We sincerely thank the reviewer for the careful reading and making the useful comments.

1-a. The quality of each experimental result is generally good, but there is little explanation or discussion about the results. Therefore, the reviewer recommends the authors to revise the manuscript and refer to the literature to add a more detailed explanation of the results.

Response: Thanks for the reviewer's careful reading and helpful suggestion. First of all, thank you for your affirmation of our work. And then thank you also for pointing out the deficiency of our manuscript and making suggestions for us. We have tried our best to improve the manuscript quality according to the concerns raised by reviewers. After re-learning the literature and supplementing the relevant experiments, we have added the related discussion and improved the manuscript, the corresponding changes in the manuscript have been marked in red. In the end, we thank you again for your kind comments.

Reviewer 2 Report

The authors describe the use of Fe3O4 nanorings as Fenton catalysts for the degradation of organic dyes.

The article is interesting and deserves to be published after some improvements.

Indeed, authors should pay attention on different aspects:

l.51 : "catalyzers" should be replaced by "catalysts"

1- The compounds should be named according to IUPAC and not commercial name. 

l.70 "Ammounium" should be corrected. phosphate monobasic = dihydrogenophosphate.

l.71: anhydrous should be at the beginning and not at the end.

2- Part 2.2

l.75: "single-crystal" is not appropriate. "crystalline" is better.

3- Part 2.5

A blank measurement should be done.

4- Part 3

All the characterizations should be linked to other references concerning the same type of species and/on measurements. It is not the first time that such measurements are done.

5- The activity of such objects should be compared to bulk Fe3O4 and other references with similar catalytic activities should be added. 

6- The high activity could be linked to higher surface area, but this parameter is not indicated, even not discussed. This is also linked to the comment 5.

In SI, Table S1 column of MB: Spaces are missing between R2 and qm.

All this has to be corrected before re-reviewing.

Author Response

Comments to the Author

The authors describe the use of Fe3O4 nanorings as Fenton catalysts for the degradation of organic dyes. The article is interesting and deserves to be published after some improvements. Indeed, authors should pay attention on different aspects. All this has to be corrected before re-reviewing.

Response: We sincerely thank the you for your kind suggestion and the valuable comments. According to reviewer's comments, we have made the related modification in our manuscript and marked them in red.

2-al.51 : "catalyzers" should be replaced by "catalysts"

Response: Thanks for the reviewer's careful examination and helpful suggestion. We have replaced the "catalyzers" with "catalysts" (Page 2, line 50).

2-b. The compounds should be named according to IUPAC and not commercial name.

Response: Thanks for the reviewer's careful reading and helpful comments. We have revised the information according to IUPAC (Page 2, line 70 - 74).

2-c. l.70 "Ammounium" should be corrected. phosphate monobasic = dihydrogenophosphate.

Response: Thanks for the reviewer's valuable reminder and helpful comments. We have removed the typos and corrected it (page 3, line 69).

2-d. l.71: anhydrous should be at the beginning and not at the end. 

Response: Thanks for the reviewer's patient inspection and helpful suggestion. We have put anhydrous at the beginning according to the comments of reviewer (page 3, line 73).

2-e. 2- Part 2.2 l.75: "single-crystal" is not appropriate. "crystalline" is better.

Response: Thanks for the reviewer's helpful comments. We have replaced "single-crystal" with "crystalline" according to the suggestions of reviewer (Page 4, line 76).

2-f. 3- Part 2.5 A blank measurement should be done.

Response: Thanks for the reviewer's constructive comments. Following the comments of the reviewer, we have added a blank measurement in our revised manuscript (Page 7, lines 194 and Page S6, Figure S3).

4- Part 3 All the characterizations should be linked to other references concerning the same type of species and/on measurements. It is not the first time that such measurements are done.

Response: Thanks for the referee's careful reading and constructive suggestion. We have added the relevant references in the revise manuscript.

2-h. 5-The activity of such objects should be compared to bulk Fe3O4 and other references with similar catalytic activities should be added.

Response: Thanks for the reviewer's constructive comments. Following the comments of the reviewer, we compared the catalytic efficiency between the Fe3O4-nanorings and solid elliptic Fe3O4-nanoparticles including Type â…  (about 337 nm) and Type â…¡ (about 196 nm). Fe3O4-nanoring exhibited better degradation effect than any other solid morphology, probably due to its cavity structure and lager specific surface area. We have provided the related information in the revised manuscript (Page 8, lines 216 - 223 and Page S8, Figure S5).

2-i. The high activity could be linked to higher surface area, but this parameter is not indicated, even not discussed. This is also linked to the comment 5.

Response: Thanks for the reviewer's constructive comments. Actually, specific surface area is a very important character for catalytic reaction. Following the comments of the reviewer, we have detected the specific surface area of the Fe3O4-MNRs. We have provided the related information in the revised manuscript (Page 5, lines 145 - 146 and Page S8, Figure S6).

2-j. In SI, Table S1 column of MB: Spaces are missing between R2 and qm.

Response: Thanks for the reviewer's careful reading and patient inspection. We have add spaces between R2 and qm according to the comments of reviewer (Page S12, Table S2).

Author Response

Comments to the Author

I therefore think the manuscript is not suited for publication in Materials, and should be greatly improved before resubmission to this or another journal. My comments follow.

Response: We sincerely thank to the reviewer for his/her careful reading and making

the useful comments.

3-a. The words "Novel" and "Regenerative" in the title are not representative of the content. Please change title. "Easily-Separated" is more representative than "Regenerative", for example.

Response: Thanks earnestly for your suggestions and comments. The title has been changed to "A Fenton-like Nanocatalyst Based on Easily-Separated Magnetic Nanoring for Oxidation and Degradation of Dyes Pollutant" (page 1, line 2 - 4).

3-b. Equations 1 - 7 are just (poorly) copied-pasted from reference 27. Almost all the reactions have errors!

(1) Not balanced, misses OH-

(2) Fe3+ instead of Fe2+ on the right side

(3) OH should be H2O

(4) Ok

(5) k misses a × 106 factor

(6) k misses a × 103 factor

(7) ok

I was quite disappointed to see such a lack of care in a submitted paper, and hope the authors will take more care revising their papers before future submission.

Response: Thanks for the reviewer's careful reading and patient inspection. We are very sorry for missing a lot factors. The missing have been amended in the main text, and we have checked the full text carefully to make sure that there are no such mistakes.

Scheme 1 is quite unusual for a scientific publication. Representing the ducks in a lake together with the iron nanorings and the H2O2is quite misleading, one could think the nanostructured catalyst and hydrogen peroxide pose no harm to animals or the environment, which is not the case.

Response: Thanks for the reviewer's valuable reminder and helpful comments. We have redrawn the Scheme 1 according to the comments of reviewer. The updated image is presented in page 3, line 66.

3-d. There is no proof in the manuscript or SI that the dyes are degraded to just H2O and CO2 (only discoloration was followed by UV Vis). This cannot actually be the case since RhB contains two nitrogen atoms, and is in the form of a chloride salt, so is methylene blue which also contains a sulfur atom, while bromophenol blue contains 4 Br atoms and a sulfur atom. One important factor to take into account is that the dye could be degraded in something more harmful for the ecosystem, even if it’s not colored anymore!

Response: Thanks for the referee's careful reading and constructive suggestion. The relevant information are modified and deleted in the Scheme and the main text. However, it was reported that dyes can be degraded to H2O, CO2 and some other substance [21, 31, 35]. Hence, we hypothesize that the dyes may be oxidized to H2O, CO2, SO2, NO2 and other non-toxic small group by Fenton-like reaction. We'll take the further investigation in our following study to find out the degradation mechanism of the other atoms such as N, S, Br, and so on.

3-e. The nanorings were already reported, and no further characterization was reported (except for the EPR in the presence of H2O2). No characterization of spent catalysts was made, despite the authors themselves state that the iron leaches under the investigated conditions.

Response: Thanks for the reviewer's frontier suggestion and constructive comments. According to the comments of reviewers, we have added the SEM, size distribution, XRD and XPS characterizations of the spent catalysts (page 9, line 230 - 233 and page S10, Figure S7).

3-f. Evidence of advantages of magnetic nanoparticles are not proven unless reference experiments are performed.

Response: Thanks for the reviewer's constructive suggestion and helpful comments. Following the suggestion of the referee, we prepared two other catalysts with different morphologies and larger size distribution. The results showed that the Fe3O4-MNRs exhibited better catalytic efficiency and reusability (Page 8, lines 216 - 223 and page S8, Figure S5).

3-g. English is poor: a lot of typos, grammar mistakes (e.g. "could recycled" instead of "could be recycled", "could rapidly" instead of "could be rapidly"), phrase construction "As the technology and society advanced, dyes are…"(wrong consecutio temporum), "Very recently, it is more appropriate" (interrupted sentence).

Response: Thanks for the referee's careful reading and helpful suggestion, we apologize for these careless mistakes. Following the suggestion of the reviewer, we have checked and corrected the spelling, expression and grammar errors throughout the updated manuscript. All changes were marked in red.

Reviewer 4 Report

In general the subject is interesting. However I’ve got impression that the Authors made some measurements like XPS and XRD just to have more graphs without even trying to squeeze more information from it. Therefore I regard the evaluation of obtained data as the main drawback of the manuscript.

Shall be the manuscript considered for publication the belowgiven remarks should be considered.

P3 – 78 After solution is dissolved completely, - unfortunate statement. Please change it appropriatly

P4 – 114 – „solutions at pH 5.0,the temperature” - space is missing

P4 – 122 – „The degradation percentage (DP)”  according to equation it is in chemical terms: well-known conversion.

P5 – description of Fig. 1 should mention about diffraction insets (Fig. 1. e, f).

154 – X-ray diffraction – Authors should try to perform measurement of average crystallite size using Scherrer equation. Based on Fig. 2 the thickness of ring wall should be about 40 nm and give rise to reflex broadening.

P6  157 – XPS analysis – This is not in-depth XPS analysis. The oxygen O1s signal should be also given. It looks like the C1s signal is pronounce. Please describe this part in more details.

General remark a.u. abbreviation is reserved for astronomical units. Not for arbitrary units.

P7 – 170 – How the UV-vis analysis was performed? In a flow of solution through spectrometer or by taking the samples?  

Author Response

Comments to the Author

In general the subject is interesting. However I’ve got impression that the Authors made some measurements like XPS and XRD just to have more graphs without even trying to squeeze more information from it. Therefore I regard the evaluation of obtained data as the main drawback of the manuscript.

Shall be the manuscript considered for publication the below given remarks should be considered.

Response: We sincerely thank to the reviewer for his/her careful reading and making

the useful comments.

4-a. P3-78 After solution is dissolved completely, - unfortunate statement. Please change it appropriatly

Response: Thanks for your suggestion sincerely. We have revised the sentence according to the suggestions of reviewer. The modified statement is as follows: The precursor solution was stirred at room temperature for 30 min. After the mixtures dissolved completely, the obtained solution was transferred into a Teflon-lined stainless-steel autoclave (page 3, line 78 - 80).

4-b. P4-114 - solutions at pH 5.0,the temperature” - space is missing.

Response: Thank you genuinely for your comments. We have added the missing space according to the suggestions of the reviewer (page 4, line 115).

4-c. P4-122- The degradation percentage (DP) according to equation it is in chemical terms: well-known conversion.

Response: Thanks for the reviewer's careful reading and helpful suggestion. We have deleted it in the revised manuscript.

4-d. P5- description of Fig. 1 should mention about diffraction insets (Fig. 1. e, f).

Response: Thanks for the reviewer's careful reading and constructive suggestion. We have described High resolution TEM and diffraction insets in detail (page 3 - 4, line 139 - 146). The modified parts are highlighted in red.

4-e. 154- X-ray diffraction - Authors should try to perform measurement of average crystallite size using Scherrer equation. Based on Fig. 2 the thickness of ring wall should be about 40 nm and give rise to reflex broadening.

Response: Thanks for the reviewer's careful reading and helpful suggestion. By learning the relevant knowledge, we know that Scherrer equation applies to uniform spherical grain and the grain sizes in 1-100 nm. According to the suggestions of reviewer, we calculate the crystallite sizes of five characteristic peaks for Fe3O4-MNRs by using Scherrer equation. The detailed values are as follows:

B obs. [2Th]

B std. [2Th]

Pos.

[2Th]

Crystallite size(Å)

0.354

0.060

30.194

280

0.394

0.060

35.593

250

0.315

0.060

43.170

335

0.315

0.060

53.644

349

0.472

0.060

57.171

219

The corresponding crystallite size of the Fe3O4-MNRs is calculated to about 22 to 35 nm by using Scherrer equation, which is near to the result of TEM (40.2 nm). The relevant details are presented in the revised manuscript (Page 5, lines 152 - 157 and Page S11, table S1).

4-f. P6 157- XPS analysis-This is not in-depth XPS analysis. The oxygen O1s signal should be also given. It looks like the C1s signal is pronounce. Please describe this part in more details.

Response: Thanks for the reviewer's frontier suggestion and constructive comments. According to the comments of reviewer, the revised details of XPS analysis was shown in the revised manuscript (Page 7, lines 172 - 177). On the other hand, the C1s signal maybe the pollution carbon in the air.

4-g. General remark a.u. abbreviation is reserved for astronomical units. Not for arbitrary units.

Response: Thanks for the reviewer's careful reading and helpful suggestion. Following the suggestion of the referees, we have redraw Figure 4 (page 6, line 170), Figure 5 (page 7, line 178) and Figure 6 (page 7, line 187) in our revised manuscript. Further, it was reported [19, 20, 35] that a.u. abbreviation is a common unit in the article and not just used in astronomy.

4-h. P7-170 - How the UV-vis analysis was performed? In a flow of solution through spectrometer or by taking the samples?

Response: Thanks for the referee's careful reading and helpful suggestion. We have added experimental details according to the comments of reviewers, which are shown in part 2, 2.5 Catalytic activity measurements (page 4, line 121 - 126).

Round 2

Reviewer 2 Report

Authors took into account some of the comments and suggestions.

However, it needs to be revised again.

l.16: remove "Next". It is not necessary

l.20: after "rapidly", "be" is missing.

l.29: "it's" should be changed by "there is"

l.30: instead of "problem", write "demand"

l.33: remove "etc"

l.70: write "ammonium"

l.71: replace "Chloride" by "chloride"

l.115: remove "Further"

l.118: remove "What's more"

l.120: add "also" after "was"

l.126: replace "liltte" by "little"

l.137: remove "And"

l.144: replace "shown" by "shows"

l.145: replace "single-crystal" by "crystalline"

A comment: Even if some crystalline areas are seen, it does not mean that the complete material is crystalline

l.157: The type I and Type II Fe3O4 nanoparticles are mentioned here but there is nothing indicating how it was done (except in SI). Add the link.

l.211-215: Indeed the concentration of the dyes has an influence on the degradation percentage. It has to be linked to the kinetics. Add some comments on it. What if the reaction time is longer ?

l.221: replace "lager" by "larger"

l.244:add "be" after "could

SI: Please reshape it since it is not easy to read as it is.

All this is necessary before being publishable

Author Response

Response to Reviewer 2:

Comments to the Author

Authors took into account some of the comments and suggestions. However, it needs to be revised again. 

Response: We sincerely thank the reviewer for the careful reading and making the valuable comments.

a.16: remove "Next". It is not necessary

l.20: after "rapidly", "be" is missing.

l.29: "it's" should be changed by "there is"

l.30: instead of "problem", write "demand"

l.33: remove "etc"

l.70: write "ammonium"

l.71: replace "Chloride" by "chloride"

l.115: remove "Further"

l.118: remove "What's more"

l.120: add "also" after "was"

l.126: replace "liltte" by "little"

l.137: remove "And"

l.144: replace "shown" by "shows"

l.221: replace "lager" by "larger"

l.244: add "be" after "could

Response: Special thanks to your good comments. Following the suggestion of the reviewer, we have revised the whole manuscript carefully and tried to avoid any grammar or syntax error. All changes were marked in red.  

2-b. A comment: Even if some crystalline areas are seen, it does not mean that the complete material is crystalline

Response: Thanks for your kind remainder and comments. In the revised manuscript, We removed the description of crystalline.

2-c. l.157: The type I and Type II Fe3O4 nanoparticles are mentioned here but there is nothing indicating how it was done (except in SI). Add the link.

Response: Thanks for the reviewer's valuable reminder and helpful suggestion. We have added relevant descriptions and reference (Page 3, line 87-94).

2-d. l.211-215: Indeed the concentration of the dyes has an influence on the degradation percentage. It has to be linked to the kinetics. Add some comments on it. What if the reaction time is longer ?

Response: Thanks for the reviewer's valuable reminder and helpful suggestion. We added the analysis in our revised manuscript. The kinetics of dyes disappearance are given in Figure S8. It can be seen that the quantities of dyes degradation increased with time.

2-e. SI: Please reshape it since it is not easy to read as it is.

Response: Thanks for your kind remainder and helpful comments. We has modified the SI so that it’s easy to read. 

Reviewer 3 Report

The authors successfully addressed most of my previous comments and as a result the manuscript is now much improved.

Nonetheless, I can not recommend publication unless the products of the reactions are analyzed and quantified. Just assuming that water and CO2 are the products of the reaction is not adequate, especially because the main message of the paper is about the functionality (i.e. catalytic activity) of a previously reported material.

I therefore recommend that the authors carry out GC on the gaseous products and analysis of the liquid by GC or HPLC, or NMR, to identify the products of the reaction. Such matter cannot be postponed to further studies since it is relevant to this particular story.

Author Response

Response to Reviewer 3:

Comments to the Author

The authors successfully addressed most of my previous comments and as a result the manuscript is now much improved.

Nonetheless, I can not recommend publication unless the products of the reactions are analyzed and quantified. Just assuming that water and CO2 are the products of the reaction is not adequate, especially because the main message of the paper is about the functionality (i.e. catalytic activity) of a previously reported material.

I therefore recommend that the authors carry out GC on the gaseous products and analysis of the liquid by GC or HPLC, or NMR, to identify the products of the reaction. Such matter cannot be postponed to further studies since it is relevant to this particular story.

Response: Special thanks to your kind comments. Indeed, it should detect all details about the dyes degradation. According to the your good instruction, we have added relevant experimental data about the degradation in our revised manuscript (Page 4, line 135-140 and Page 9-10, line 249-260). We taken a gas chromatography-mass spectrometry (GC-MS) to identify the products of the reaction. Briefly, after 60 min reaction, for the MB solution, some small organic molecules such as methylbenzene, styrene, phenol, p-nitrophenol, trimethylphenol and 4-Aminothiophenol, N, N, S-trimethyl can be detected (Figure 10a). For the RhB solution, some small organic molecules such as methylbenzene, phenol, p-nitrophenol and trimethylphenol can be detected (Figure 10b). For the BPB solution, some small organic molecules such as tetrahydro-2,5-dimethyl, methylbenzene, phenol and trimethylphenol can be detected (Figure 10c). After 120 min degradation, the small organic molecules were degraded as CO2 (Figure S9) and inorganic ions (Figure 10d, e and f). For the MB, the final products are SO42−, NH4+ and NO3, respectively. For the RhB, the final products are possibly NH4+ and NO3. For the BPB, the final products are Br and SO42−. Accordingly, the dyes were degraded.

Reviewer 4 Report

The Authors have addressed all my remarks in satisfactory manner. 

Therefore I recommend this manuscript for publication. 

Author Response

Response to Reviewer 4:

Comments to the Author

The Authors have addressed all my remarks in satisfactory manner. Therefore I recommend this manuscript for publication.

Response: Special thanks to your kind comments. We’ve done a thorough inspection of the revised manuscript.

Round 3

Reviewer 3 Report

The manuscript is now much improved and I am pleased to see that thanks to the revision the authors can now show what exactly is produced, without just assuming total degradation, which indeed is not the case.

I recommend the manuscript to be accepted for publication on Materials after minor revision, that is, after including a small comment about further research needed to lead to complete degradation, due to the possible (or known) toxicity of the by-products.

Author Response

Response to Reviewer 3:

Comments to the Author

The manuscript is now much improved and I am pleased to see that thanks to the revision the authors can now show what exactly is produced, without just assuming total degradation, which indeed is not the case.

I recommend the manuscript to be accepted for publication on Materials after minor revision, that is, after including a small comment about further research needed to lead to complete degradation, due to the possible (or known) toxicity of the by-products.

Response: We sincerely thank you for the careful reading and making the valuable comments. We have provided the relateddiscussion in the Conclusions Section (Page 10-11, lines 273-276).